

# A Comparison of Lossless Compression Algorithms for Altimeter Data

Mathieu Thevenin[1], Stephane Pigoury[2], Olivier Thomine[3], and Flavien Gouillon[4]

[1]CEA SPEC, Bat 772 F91191 Gif-sur-Yvette - France
[2]Subnet SAS, 21 av. de la belle image, 94440 Marolles-en-Brie - France
[3]Insa Rouen Normandie, 685 Av. de l'Université, 76800 Saint-Étienne-du-Rouvray - France
[4]CNES, Centre Spatial de Toulouse, 18 avenue Edouard Belin, 31401 Toulouse, France

**Correspondence:** Mathieu Thevenin mathieu.thevenin@cea.fr

**Abstract.** Satellite data transmission is usually limited between hundreds of kilobits-per-second (kb/s) and several megabits-per-second (Mb/s) while the space-to-ground data volume is becoming larger as the resolution of the instruments increases while the bandwidth remains limited, typically. The Surface Water and Ocean Topography (SWOT) altimetry mission is a partnership between the National Aeronautics and Space Administration (NASA) and the Centre National des Études Spatiales (CNES) which uses the innovative KaRin instrument, a $K_a$ band (35.75 GHz) synthetic aperture radar combined with an interforemeter. Its launch is expected for 2022 for oceanographic and hydrological levels measurement and it will generate 7 TeraBytes-per-day, for a lifetime total of 20 PetaBytes. That is why data compression needs to be implemented at both ends of satellite communications. This study compares the compression results obtained with 672 algorithms, mostly based on the Huffman coding approach which constitute the state-of-the-art for scientific data manipulation, including Computational Fluid Dynamics (CFD). We also have incorporated data preprocessing such as shuffle and bitshuffle, and a novel algorithm known as *SL6*.

## 1 Introduction

Satellites for Earth observations is a topic for research and development, which has been historically taken-on by academia and institutions. More recently by private operator are working on this topics. The observation devices are based on different technologies such as visible light collection, radar and hyperspectral imaging, interferometry. These technologies can be combined to be used together. This, in association with an increasing number of satellites in service, has led to an important rise in the amount of data collected that has to be transmitted to the ground, which is then processed and stored in a database for further use (Sudmanns et al. (2020)). Traditionally, satellite data transmission relies on radio-frequencies (Elsey (1968)). Data may transit on Data Relay Satellites (DRS) like in International Telecommunication Union (2017); Radhakrishnan et al. (2016) or can be directly sent to ground datacenters or to terminals (Fraire et al. (2019)) in unidirectional or bidirectional manners, depending on the application. Despite the most recent advances, the available bandwidth remains limited; indeed, lightweight terminals only have a few hundred of kilobits-per-second ($kb/s$) capacity (WMO (2018)). Of course, more important communication links can reach a few Megabits-per-second ($Mb/s$) (under 3 GHz frequency bands) to a few hundreds of $Mb/s$ for



the inter-orbital datalinks (usually over 10 GHz frequency bands). In some cases, the satellite has a limited window of time to
transmit the data to the ground – *i.e.* where it is aligned with the reception antenna. Data compression needs to be used and
implemented at both ends of the transmission devices (space and ground). The compression scheme is usually application and
data dependent like in Huang (2011), in observation, the biggest volume of data is transmitted from the space to the ground.

The Surface Water and Ocean Topography (SWOT) (Vaze et al. (2018)) mission is a partnership between the National Aero-
nautics and Space Administration (NASA) and the Centre National des Études Spatiales (CNES), and continues the long history
of altimetry missions with an innovative instrument known as KaRin (Fjørtoft et al. (2014)), which is a $K_a$ band (35.75 GHz)
synthetic aperture radar associated to an interferometer as illustrated by the Figure 1. The SWOT mission launch is foreseen for
year 2022 and addresses both oceanographic and hydrological communities. It aims at accurately measuring the water level of
the oceans, the rivers and the lakes. It is expected that the SWOT mission will generate about 20 Petabyte (PB) of data during
the mission lifetime which corresponds approximately over 7 TB-per-day. Even if the data format is not fully defined yet for
the SWOT mission, the kind of data generated by such missions is usually stored in Hierarchical Data Format 5 (HDF5) (Trott
et al. (1996)) files. The data volume issue has been addressed by the implementation of different compression schemes (De-
varajan et al. (2019); Welton et al. (2011)) in the tools for the manipulation of the HDF5 formats of the Computational Fluid
Dynamics (CFD) General Notation System (CGNS) and benchmarked by previous works like in Delaunay et al. (2019); Di and
Cappello (2018). The most recent pone shows that the combination of shuffle preprocessing and deflate lossless compression
(level 4) provides good results. Moreover, it states that in the case of using a lossy compression, a required precision must be
defined by the scientists as a Number of Significant Digits (NSD) for each dataset variable. Naturally, when using lossy com-
pression algorithms, the full precision of the data is not always available, which can be problematic for a scientific use of the
data. In return of the lost of precision, lossy algorithms generally have a better compression rate than the lossless ones. Since
scientific data needs to keep the original precision, lossless compression algorithms are preferable when possible. A key point
that need to be evaluated is the consistency of the compression level depending on the nature of the data and the homogeneity
of compression time and throughput. Indeed, variability in the compression rates leads to non deterministic results when it
comes to data transfer, especially for a limited time frame.

This paper proposes to investigate different lossless compression algorithms for scientific data, – especially in CFD – that
are representative of the usual earth observation data. This article extends the work presented in Delaunay et al. (2019) by
exploring the algorithms that are traditionally used in HDF5 and by adding another one to the benchmark.

If the compression level is an important metric, it does not provide any information on the potential benefit of on-the-fly
compression/decompression during data processing. That is why key points and the use of specific metrics for evaluating the
data compression algorithms performances will be investigated:

– compression level defined here by $C_r$;

– compression throughput;

– the homogeneity of the compression regarding the fields of different types (float, integer) and nature;

– the memory usage:





**Figure 1.** Illustration of the SWOT Ka-band Radar interferometer (KaRin) and nadir altimeter measurement concept and the products generated (blue and red shapes on the ground) inspired from Vaze et al. (2018).



– the ability to compress data on-the-fly, in other words, does it needs the storage of the full data in memory;

The major contributions of this paper are: a) the proposal of a thorough bench methodology for lossless compression algo-
rithm for scientific data; b) the proposal of metrics that goes beyond the compression rate; c) a selection of lossless compression
algorithms suitable for advanced CFD data and d) discussion of different scenarios that can benefit from compression algo-
rithms in the domain of the earth observation.

This paper is organized in five sections. The first one is this introduction. The second one depicts the methodology followed
for this research. The third section depicts the results and focuses on the most interesting algorithms. The fourth section
discusses the way how the most efficient algorithms can be used in different environments and infrastructures, including High-
Performance Computing (HPC) and embedded systems. Finally, the last section provides the conclusions of this research.

## 2 Methodology

The section describes the methodology developed in this study. It is made of four subsections: first it starts with a description
of the dataset; followed by a description of the metrics that allow us to choose the compression algorithms that perform the
best. Finally, the testbench and data management approach is explained followed by the results we have obtained.

### 2.1 Dataset

The scientific community relies on CGNS which is a binary unstructured hierarchical format (Diane Poirie and et al. (1998);
Christopher Rumsey, Bruce Wedan and Poinot (2012)) implementing Advanced Data Format (ADF) (Owen and Daniel (1998))
and HDF5 (Folk et al. (2011)) to store observational data. It aims at providing a standard for recording and recovering computer
data associated to the numerical solution of the equations of fluid dynamics. It also implements shapes, up to three dimensions:
0-D point; 1-D line; 2-D triangle and quadrangle; 3-D tetrahedron, pyramid, pentahedron, hexahedron. Due to the data volume,
not only the transmission and storage are problematic, the access and the read/write time are significant bottlenecks for both
post-processing and simulations (Soumagne et al. (2010)).

The dataset used for this study corresponds to the SWOT mission products. The SWOT mission will generate two types of
products: the high-resolution products, which are dedicated to the hydrology thematic, and the low-resolution products, which
are mostly dedicated to the oceanography domain. Basically, L0 data contain raw telemetry; L1 Single Look complex means
that each pixel encodes its magnitude (I and Q) and therefore contains both amplitude and phase information. Each I and Q
value is encoded using 16 bits per pixel but stored in 32-bit floating point datawords. The Pixel Cloud product (L2_HR_PIXC)
contains data from the KaRin instrument configured in High Resolution (HR) mode – *i.e.* the HR mask is enabled; it corre-
sponds to the pixels that are detected as being over water. The "Pixel Cloud product" is organized into sub-orbit tiles for each
swath and each pass, and this is an intermediate product between the L1 Single Look Complex products and the L2 lake/river
ones. As illustrated in Figure 1, the product granularity is $64\ km \times 60\ km$ tile.

The dataset used for this study is a simulated golden dataset generated using data obtained from the SWOT mission and
although, it might not be the most recent, it is based on the one used in Delaunay et al. (2019); for better comparisons. It





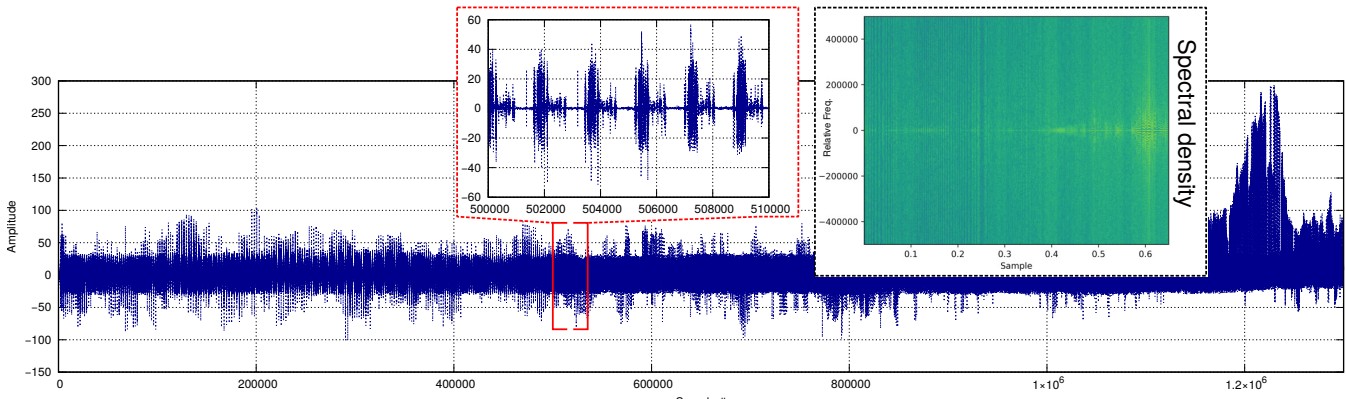

**Figure 2.** Illustration of the samples of the `Pixel Cloud Height` File (64 bits Float), the signal appears to be periodic but with quite chaotic characteristics.

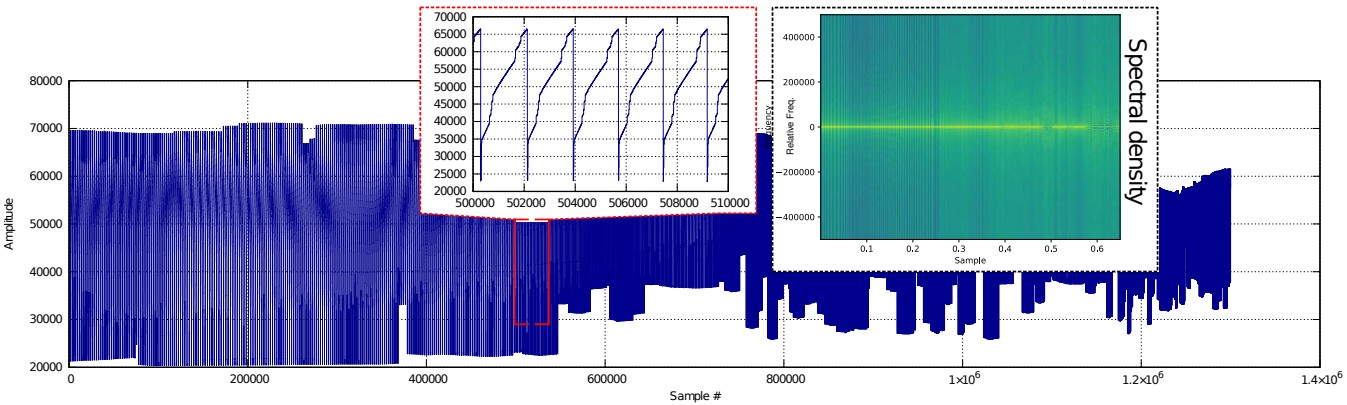

**Figure 3.** Illustration of the samples of the `Pixel Cloud L2H CrossTrack` File (64 bits Float), the signal seems to be locally continuous and periodic.

consists of four HDF5 files. Two of them contain the measurements and the two others contains synthetic generatedd data divided into fields as follow:

- *signal 1 (s1)*: synthetic data that contains $106\,954\,752$ samples of little-endian (LE) 32-bit floating-point (FP) ($427.8$ MB), its name is : `s1: signal`;

- *signal 3D (s3D)*: synthetic data, that contains $1\,048\,576$ samples of LE 32-bit FP ($4.2$ MB), the related field is `s3D: signal`;





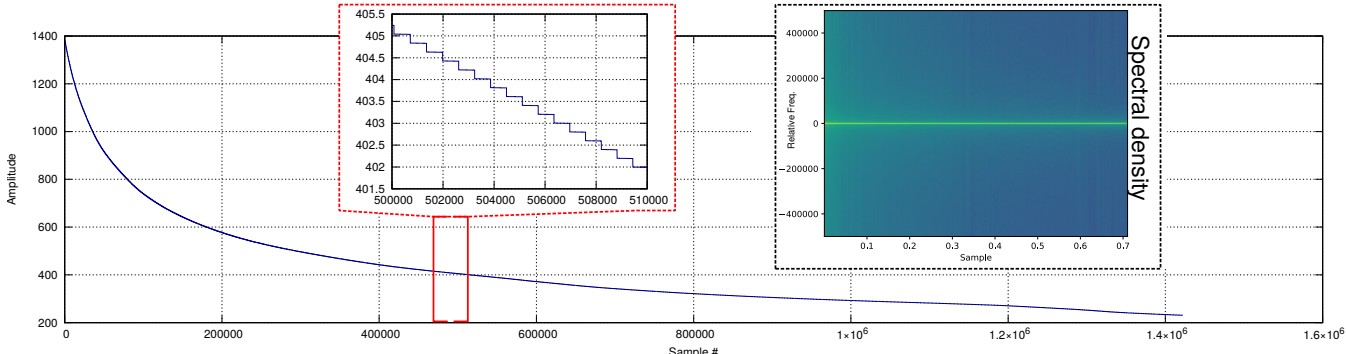

**Figure 4.** Illustration of the samples of the `swot pixel_area` file (64 bits Float), the data of this file are continuous, a good compression ratio is expected.

- 12 fields of real world measurement data, that contain $1\,300\,111$ samples of LE 64-bit FP (10.4 MB), their names are `pixel_cloud`: `classification`, `coherent_power`, `cross_track`, `dheight_dphase`, `dlatitude_dphase`, `dlongitude_dphase`, `height`, `illumination_time`, `incidence_angle`, `latitude`, `longitude`, `pixel_area`;

- 7 fields of experimental data that contain $1\,421\,888$ samples of LE 64-bit FP (11.4 MB), their names are `SWOT_L2`: `cross_track`, `height`, `illumination_time`, `latitude`, `longitude`, `pixel_area`, `range_index`;

- 9 fields of experimental data that contains $1\,300\,111$ samples of LE 32-bit FP (5.2 MB), their name are `pixel_cloud`: `continuous_classification`, `num_med_looks`, `num_rare_looks`, `phase_noise_std`, `power_left`, `power_right`, `sigma0`, `x_factor_left`, `x_factor_right`;

- 1 field of experimental data that contains $2\,600\,222$ samples in two dimension of LE 32-bit FP (10.4 Mb), its name is `pixel_cloud`: `ifgram`.

In order to estimate the compression performance on each type of data, every single field had been extracted and compression/decompression was performed on each of them in the testbench. The Figures 2, 3 and 4 illustrates how different the fields are from each other (extracted from the SWOT file). Figure 2 shows a high entropy and low correlation between the samples, while Figure 3 shows a high correlation and quite high entropy in the signal. Finally, Figure 4 shows high correlation and a low entropy signal. Field extraction is performed using the HDFtools and generates a binary file encoded in the original format of the related field.

## 2.2 Lossless Compression Algorithms

Several compression approaches have been proposed through the past decades in the domain of earth observation and scientific data. HDF5 data representation is widely used in the community, thus naturally came the need of compressing this file structure.



Within years, CFD tools, that are use to produce the HDF5, have started to implement compression and decompression algorithms. Many different approaches are proposed for data compression in Jayasankar et al. (2021). Today's state-of-the-art works revolve around the use of Huffman, entropy encoders, artihmetic encoders, they are implemented in LZ, LZ4 etc. algorithms. They consist of building a dictionary of redundancies in the signal, then, only the dictionary and the indexes of the words are transmitted. By themselves, these approaches perform well, but they require the analysis of huge portions of the data before

being able to start the compression. Moreover, they are more well-suited for strings than floating-point encoded samples. For this reason, some works focused on using data preconditioners to reorganize the data. Shuffle, for example, analyses the entire file or chunks of data, to reorganize the values to get consecutive similar-valued samples, this way, it permits to the encoders compressor to perform well. Among the most recent advances, a bit-level filter was proposed in Masui (2017), initially associated with a lossy compression algorithm (Masui et al. (2015)), it has been demonstrated that it can be efficiently combined

with lossless ones as given in Delaunay et al. (2019). Because of their impact on the compression level when combined with Huffman or entropy encoders; the shuffle and the bitshuffle preconditioners are implemented in the netCFD tools.

Delta coding is a basic approach that consists of encoding the derivative of the signal. This way, all the values are zero-centered, and most of the bits of the encoded samples are set to zero. This approach is lossless on integer encoded, but lossy when used on floating-point encoded data, even if the loss is limited. Recently, a variant inspired by delta-coding was proposed

in the domain of the HPC (Lloyd et al. (2018)) to reduce the required bandwidth for data transmission between nodes. Thanks to a parallel implementation, it sometimes outperforms the other approaches. But again, LZ4fast performs better in many situations. The delta coding approach can also be combined as done in Patauner et al. (2011) where Huffman coding and delta coding are used together. If the traditional delta coding is not really efficient for floating point coded samples, and more generally inferior to the traditional lossless compression schemes , a more recent approach, named here *SL6* (Thomine et al.

(2016, US Patent 20,190,044,532)), that consists in encoding the difference between the signal and an approximated value makes possible to slightly reduce the number of bits used to transmit the signal. A whole compression and decompression framework is available for *SL6* as well in hardware and software under commercial license.

To summarize, the Huffman-based entropy-encoders and arithmetic-encoders based algorithms will be compared to the recent class of algorithms known as *SL6* (Thomine et al. (2016, US Patent 20,190,044,532)). The comparison will be done on

the SWOT dataset which contains integer and floating point number. The shuffle and bitshuffle preconditioners will be used for the LZ family algorithms, as it has been shown in the past that they help them to compress integer and floating-point samples.

## 2.3   The Testbench Process

Since it provides most of today's compression algorithms, the benchmark we developed is derived from the opensource *lzbench* as it was done in Kunkel (2017). Originally, *lzbench* focuses on the LZ77/LZSS/LZMA compression algorithms and performs

in-memory, thus the results are independent from the disk reading and writing times. All the compression algorithms are compiled from sources. This way they all use the same compiler and the same options. Files are compressed and decompressed and compared to the original ones, the time is measured using C primitives. A total of 52 LZ-family algorithms were tested,the LZ family algorithms provided by the *lzbench* are: *blosclz, brieflz, brotli, crush, csc, density, fastlz, gipfeli, libdeflate, lizard, LZ,*





*lz4, lz4fast, lz4hc, lzf, lzfse, lzg, lzham, lzjb, lzlib, lzma, lzmat, lzo1, lzo1a, lzo1b, lzo1c, lzo1f, lzo1x, lzo1y, lzo1z, lzo2a, lzrw,*
*lzsse2, lzsse4, lzsse8, lzvn, memcpy, pithy, quicklz, lzfse, lzg, lzham, lzjb, lzlib, lzma, lzmat, lzo1, lzo1a, lzo1b, lzo1c, lzo1f, lzo1x,*
*lzo1y, lzo1z, lzo2a, lzrw, lzsse2, lzsse4, lzsse8, lzvn, pithy, quicklz, shrinker, slz_zlib, snappy, ucl_nrv2b, ucl_nrv2d, ucl_nrv2e,*
*wflz, xpack, xz, yalz77, yappy, zlib, zling and zstd.* The version of these algorithms are the latest available version. For each of
them, the most common variants were tested. The variants are usually identified using one or several parameters.

The *SL6* algorithm is also benched, again using different parameters, originally designed for the compression of messages
between nodes of HPC clusters, its lightweight properties and low computing and memory requirements are properties that
make it interesting for the SWOT mission. The parameters are the size of the block and the interpolation mode as explained
in Thomine et al. (2016, US Patent 20,190,044,532). Moreover, we added the shuffle and bitshuffle preconditioners. This lead
to a total of 672 different versions.

The results are compared to the in-memory data copy *memcpy* which provides no compression but maximum throughput –
4.2 GB/s measured

The workstation used to execute the testbench is an Intel(R) Xeon(R) CPU E5-2620 v3 running at $2.40$ GHz processor, $64$ GB
of memory. Since this study aims at selecting the algorithms that could be embedded on an onboard computer for a mission, the
multicore capacities of the architecture were disabled. Other compression algorithms are excluded from this study. For example
the case of *Zarr* as it is designed to be directly used in Python code, and thus cannot not provide a throughput comparable to
the others. *Blosc* (Howison (2013)) is a compression schemes that relies on other compression codecs. It implements fast
data accesses to exploit the processor cache and Single Instruction Multiple Data (SIMD) instruction (SSE, Altivec *etc.*) and
implements shuffle and bitshuffle filters. The compression itself is performed using external codecs, usually *FastLZ*; thus,
potentially, any compression algorithm could be used with *Blosc*. This is not evaluated here since it would be a step forward.
Another key point is that the algorithms we are considering here are available as C libraries or source code and thus can be
suitable both for HPC or embedded applications.

## 2.4 Definition of the Metrics

To ensure a fair comparison of the different algorithms, several metrics are considered, including metrics we specifically propose in this paper. First, the ability to compress, expressed in percent, is calculated by $C_r = \frac{size_{orig} - size_{comp}}{size_{orig}} * 100$ where
$size_{comp}$ refers to the size of the file after compression and $size_{orig}$ refers to the size of the algorithm before compression.
From this measurement, one can calculate the mean ($M$) and the standard deviation ($\sigma$) for a set of fields, which also allows to derivate the coefficient of variation calculated by $\widehat{C_v} = \sigma/M$. Secondly, the compression throughput $C_{throughput}$, or
compression speed, is calculated using the size of the original data divided by time required by the algorithm to fully perform compression: $C_{throughput} = size_{orig}/time_{comp}$. This is the same approach for the decompression with $D_{throughput} = size_{orig}/time_{decomp}$. Here again, the corresponding $C_v$ can be calculated from the standard deviation $\sigma$.

As one of the targeted application is the on-board compression and data transmission, it is important to put emphasis on
the algorithms with consistent performances, whatever the type of data. In other words, an algorithm providing extremely
different compression $C_r$ and/or times, depending on the nature of the data, should get a lower score than an algorithm which





has consistent results among the whole dataset. Intuitively, it means the highest throughput and the highest compression $C_r$ are preferable but with the lowest standard deviation for both of them. Thus, we propose the metric $H$-score for a given set of data (fields) calculated on the normalized results (maximum set to 1) using the Equation 1, with, $\sigma(C_r)$ the standard deviation of the compression rate, $T$ the throughput and $\sigma(T)$ its the standard deviation. A graphic intuitive illustration of the variables used in the $H$-score is given in Figure 5. Moreover, we defined $\alpha$ and $\beta$ coefficients, which are set to one. They can be used to weight either the compression $C_r$ ($\alpha$) or the throughput ($\beta$). Finally, the $\rho$ coefficient, set to 1000, is simply used to ease the reading of the results. A low or negative $H$-score means the algorithms results are heterogeneous within the selected data, on the opposite side, a higher $H$-score means the algorithms results are homogeneous among the dataset while providing good performances.

$$H-score = \underbrace{\rho}_{1000} \times (\overline{C_r} - \underbrace{\alpha}_{1} \cdot \sigma(rate)) \cdot (\overline{T} - \underbrace{\beta}_{1} \cdot \sigma(T)) \tag{1}$$

The third metric that was considered in this paper, and proposed by Thomine, is called the $tt_{tx}$ for transmission-throughput-threshold and is expressed in Bytes-per-second. It is used to determine if it is worth compressing the data when a certain transfer $C_r$ is available for data transmission. In other words, if $tt_{tx}$ is higher than the media throughput (or speed), then the time used to compress the data, to transmit and to decompress it, is lower than transmitting the original data; in other words, the higher $tt_{tx}$ is better. We can derive a similar metric for writing the data to a memory or to a disk. In this case, as decompression does not need to be performed, its time is removed from the calculation; it gives $tt_{wr}$ which stands for writing-throughput-threshold, also expressed in bits-per-second. $tt_{tx}$ and $tt_{wr}$ are given in the Equations 2 and 3, $Size_{orig}$ refers to the original size of the data, $Size_{comp}$ the size of the compressed data, $Time_{comp}$ and $Time_{decomp}$ the time required to compress and decompress it.

$$tt_{tx} = \frac{Size_{orig} - Size_{comp}}{time_{comp} + Time_{decomp}} \tag{2}$$

$$tt_{wr} = \frac{Size_{orig} - Size_{comp}}{Time_{comp}} \tag{3}$$

## 2.5 Data management

The number of algorithms is quite important and for each algorithm, many variation are considered. For example, the impact of the algorithms parameters (more than 17 variants for ZStd), but also, the use of preconditioning filters such as shuffle and bitshuffle. Since the volume of data produced by our extended testbench is quite important we decided to store all the results in a relational database we have specifically designed for the analysis of the results. This way, all the data can be extracted and analyzed using a combination of SQL queries. In the same vein, the derivated metrics such as compression $C_r$, compression and decompression throughput, ($tt_{tx}$ and $tt_{wr}$ can be calculated on-the-fly directly in the queries. Moreover, to make it easier, we have designed a graphical user interface (Figure 7) to browse through the results, it allows an instant displaying, filtering,





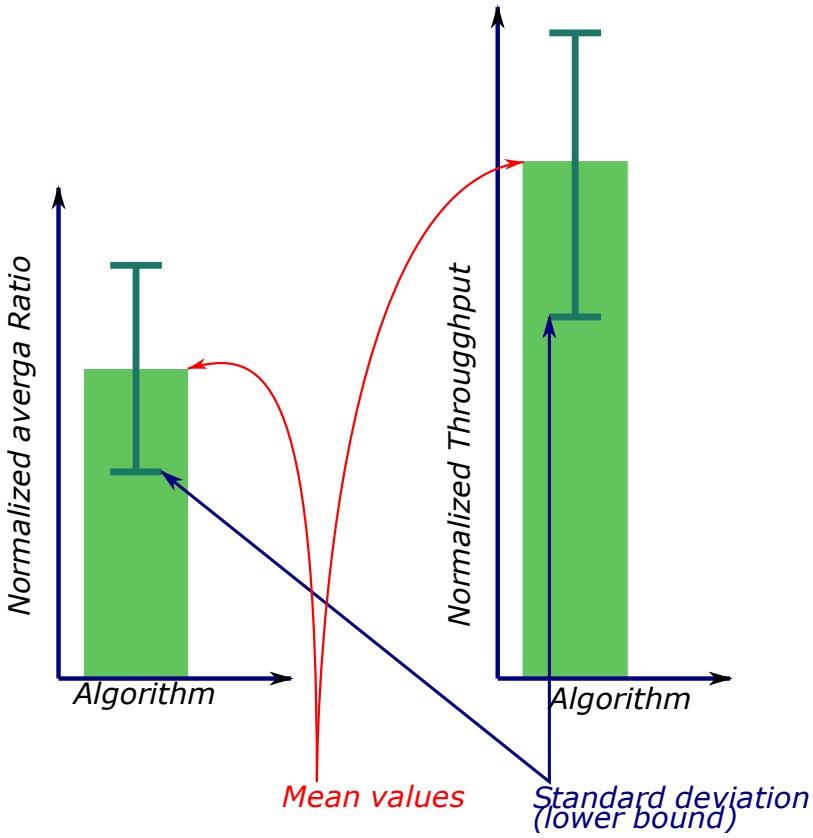

**Figure 5.** Illustration of the variables used to calculate the $H$-score; intuitively, the lower-bound of error bars made out of the standard deviation ($\sigma$) is subtracted from the mean compression $C_r$ and speeds for a given dataset. The result of the subtraction is multiplied. Coefficients ($\alpha$, $\beta$) are added to weight the $\sigma$ values (set to 1) as given in the Equation 1.

ranking and analysis of the different metrics used. This user interface was designed for internal use only, but can be distributed on demand.

## 3 Results obtained using the Testbench

This section presents the results we have obtained with the test-bench. First, a general overview of the results is introduced; then, we focus on the fields that appear to be difficult to compress.

### 3.1 General Results

As all 672 variants of the algorithms we have benchmarked cannot be presented in a concise way, we choose to focus on the most interesting of them, based on the results obtained using the metrics defined in the previous section:



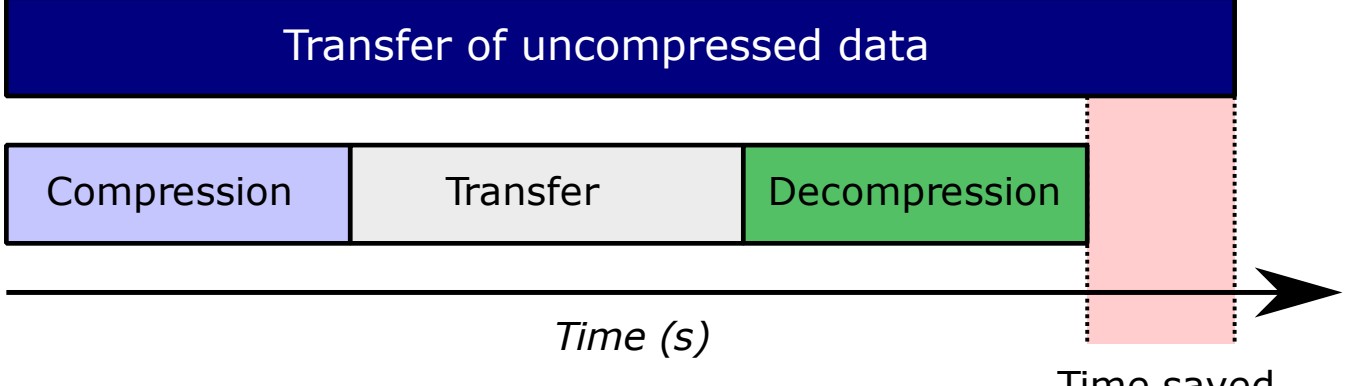

**Figure 6.** illustration of the time saved by compressing, transferring and decompressing; the calculation of the $tt_{tx}$ is used to emphasize the threshold from which time is saved; when it comes to data writing into a memory or to a disk, $tt_{wr}$ is used, the decompression time can be omitted since it is supposed to be done at the reading process.

- – `lz4fast` associated to shuffle, parameter 17;

- – `LZ4fast` associated to bitshuffle, parameter 17;

- – `Zstd` parameter 22;

- – `Zstd` associated to shuffle, parameter 1;

– `Zstd` associated to bitshuffle, parameter 1;

- – `LZ` associated to shuffle, parameter 9;

- – `LZ` associated to bitshuffle, parameter 9;

- – *SL6* with a polynomial sample estimator.

The measured time resolution on the workstation devoted to the bench is 1.7 $\mu$s $\pm$ 0.4 $\mu$s. Table 1 provides the results
for the `S1` fields of the signal file. It shows a certain variability in the compression $C_r$ (mean = 63.34 %; $\sigma = 2.36$) and a higher one in the compression and decompression speeds: $mean = 559\,263\,973; \sigma = 477\,136\,146.9$ for the compression and $mean = 718\,261\,809\,; \sigma = 513\,374\,554$ for the decompression. One can see the standard deviation on the compression speed is extremely high and close to the mean value.

The results obtained on the SWOT `pixel_cloud` file are quite different in terms of ranking, due to the nature of
the data which is slightly different. They are given in Table 2 where we can see, again, an homogeneous compression $C_r$ ($mean = 54.21; \sigma = 3.65$); heterogeneous compression speed ($mean = 383\,153\,349; \sigma = 313\,197\,839$). This is the same for the decompression speed with ($mean = 652\,996\,328; \sigma = 509\,295\,674$).





**Figure 7.** screenshot of the graphical user interface designed for the analysis of the data produced by the testbench; a SQL database is used to store the results.

The entropy level associated to the fields is higher and the fields are more complex to compress using the traditional Huffman-based approaches. The LZ4Fast and LZ associated to the shuffle filter obtain good results, but *SL6*, which exploits the correla-
tion between samples scores higher in terms of throughput with a compression $C_r$ close to the others.

A similar behavior is observed on the SWOT L2HR file, as given in Table 3: the compression $C_r$ are quite homogeneous $\sigma = 0.95$ while a great variability appears in the compression and decompression speed with a $\sigma = 639\,050\,509$, which is close to the mean value $680\,465\,679$ for compression speed; $\sigma = 825\,523\,818$ for a mean value of $849\,856\,905$ for decompression speed. It means that the throughput highly depends on the algorithms and on the nature of the data.

However, the ranking is different, ZStd with no filter provides the highest throughput, but lowest compression $C_r$. Closely followed by *SL6* which has a higher (among the highest) compression $C_r$ and is faster at decompressing. The other algorithms are slightly slower as the compression $C_r$ are similar or a few percent lower than the best ones.



**Table 1.** Summary of the results obtained for the S1 signal file, original size 8 388 608 B.

| File | Par. | Preproc. | $C_r$ | comp. time | Decomp. time | Comp. speed | Decomp. speed | Comp. size |
| --- | --- | --- | --- | --- | --- | --- | --- | --- |
| | | | (%) | (ns) | (ns) | (Bytes/s) | (Bytes/s ) | (Bytes) |
| ZSTD | 1 | none | 58.50 | 10 477 144 | 5 958 746 | 800 657 889 | 1 407 780 765 | 3 481 550 |
| ZSTD | 1 | shuffle | 65.68 | 10 155 354 | 8 260 379 | 826 028 123 | 1 015 523 380 | 2 878 818 |
| ZSTD | 1 | bitshuffle | 65.34 | 38 472 020 | 30 087 028 | 218 044 387 | 278 811 453 | 2 907 213 |
| LZ | 9 | none | 58.24 | 254 577 532 | 33 740 694 | 32 951 093 | 248 619 901 | 3 503 334 |
| LZ | 9 | shuffle | 66.84 | 132 626 450 | 25 893 111 | 63 249 887 | 323 970 650 | 2 781 908 |
| LZ | 9 | bitshuffle | 65.58 | 170 177 748 | 48 532 690 | 49 293 213 | 172 844 489 | 2 887 199 |
| LZ4 | 0 | shuffle | 62.15 | 7 712 785 | 7 685 319 | 1 087 623 731 | 1 091 510 710 | 3 174 678 |
| LZ4 | 0 | bitshuffle | 64.91 | 32 678 452 | 29 960 194 | 256 701 511 | 279 991 778 | 2 943 353 |
| LZ4FAST | 3 | shuffle | 62.10 | 6 445 824 | 7 242 951 | 1 301 401 962 | 1 158 175 445 | 3 178 959 |
| LZ4FAST | 3 | bitshuffle | 64.75 | 32 926 048 | 29 876 013 | 254 771 177 | 280 780 705 | 2 957 242 |
| LZ4FAST | 17 | shuffle | 62.11 | 6 516 309 | 7 435 122 | 1 287 325 079 | 1 128 240 801 | 3 178 408 |
| LZ4FAST | 17 | bitshuffle | 64.39 | 31 675 455 | 29 930 478 | 264 829 913 | 280 269 764 | 2 987 309 |
| *SL6* | Poly. | none | 63.12 | 10 572 072 | 5 160 822 | 793 468 679 | 1 625 440 288 | 3 093 395 |

**Table 2.** Summary of the results obtained for the SWOT `pixel_cloud` signal file, size 208 017 760 Bytes.

| File | Param. | Preproc. | $C_r$ | comp. time | Decomp. time | Comp. speed | Decomp. speed | Comp. size |
| --- | --- | --- | --- | --- | --- | --- | --- | --- |
| | | | (%) | (ns) | (ns) | (Bytes/s) | (Bytes/s ) | (Bytes) |
| ZSTD | 1 | shuffle | 49.39 | 444 398 317 | 249 599 814 | 468 088 541 | 833 405 108 | 105 280 832 |
| ZSTD | 1 | bitshuffle | 47.15 | 1 181 907 250 | 756 880 412 | 176 001 763 | 274 835 703 | 109 940 201 |
| LZ | 9 | none | 39.00 | 21 218 526 054 | 845 143 541 | 9 803 591 | 246 133 053 | 126 894 903 |
| LZ | 9 | shuffle | 50.46 | 21 854 440 692 | 654 839 210 | 9 518 329 | 317 662 346 | 103 043 840 |
| LZ | 9 | bitshuffle | 47.79 | 14 458 854 352 | 1 299 657 146 | 14 386 877 | 160 055 874 | 108 602 087 |
| LZ4 | 0 | shuffle | 46.00 | 291 760 551 | 205 508 459 | 712 974 250 | 1 012 210 208 | 112 324 430 |
| LZ4 | 0 | bitshuffle | 45.97 | 881 074 723 | 730 831 535 | 236 095 480 | 284 631 615 | 112 399 104 |
| LZ4FAST | 3 | shuffle | 45.77 | 291 303 202 | 201 869 174 | 714 093 627 | 1 030 458 271 | 112 805 305 |
| LZ4FAST | 3 | bitshuffle | 45.76 | 877 821 970 | 729 865 607 | 236 970 328 | 285 008 306 | 112 832 614 |
| LZ4FAST | 17 | shuffle | 44.83 | 253 447 978 | 202 266 864 | 820 751 310 | 1 028 432 220 | 114 772 156 |
| LZ4FAST | 17 | bitshuffle | 44.92 | 837 072 886 | 729 876 936 | 248 506 150 | 285 003 882 | 114 585 439 |
| *SL6* | - | none | 42.98 | 239 367 624 | 103 952 649 | 869 030 475 | 2 001 081 858 | 118 608 676 |

Unsurprisingly, it appears the set of algorithms which shows the best performance are similar to the one presented in Delaunay et al. (2019) even if the method and the test-bench differs from the one presented in this paper. The rest of the article will

focus on the SWOT fields as they seem to provide the lowest compression $C_r$ compared to the others – which compress quite well whatever algorithm is used.

### 3.1.1 Compression $C_r$

First of all, the unweighted average compression $C_r$ for the fields of the SWOT file is 74.36 %, with a $\sigma = 37.39$, the details are given in Figure 8. As LZ4 and LZ4fast are not able to compress 17 fields out of 72, they obtain the worst results with an

average compression $C_r$ of 67.22 and 66.51 ($\sigma = 45.81$ and $\sigma = 46.49$). The median compression $C_r$ for the SWOT file fields





**Table 3.** Summary of the results obtained for the SWOT `L2HR` signal file, original size $459\,269\,824$ Bytes.

| File | Param. | Preproc. | $C_r$ | comp. time | Decomp. time | Comp. speeed | Decomp. speed | Comp. size |
|------|--------|----------|-------|------------|--------------|--------------|---------------|------------|
| | | | (%) | (ns) | (ns) | (Bytes/s) | (Bytes/s) | (Bytes) |
| ZSTD | 1 | none | 89.44 | 245 576 988 | 187 199 279 | 1 870 166 369 | 2 453 373 894 | 48 500 312 |
| ZSTD | 1 | shuffle | 92.65 | 504 396 287 | 475 060 914 | 910 533 713 | 966 759 863 | 33 760 218 |
| ZSTD | 1 | bitshuffle | 92.26 | 1 833 936 246 | 1 680 779 517 | 250 428 457 | 273 248 109 | 35 545 956 |
| LZ | 9 | none | 90.51 | 7 658 339 217 | 1 170 823 423 | 59 969 898 | 392 262 245 | 43 603 304 |
| LZ | 9 | shuffle | 92.78 | 6 934 283 461 | 1 574 703 688 | 66 231 764 | 291 654 759 | 33 159 715 |
| LZ | 9 | bitshuffle | 92.46 | 8 688 358 240 | 2 503 710 877 | 52 860 369 | 183 435 647 | 34 640 701 |
| LZ4 | 0 | shuffle | 91.88 | 461 088 354 | 457 298 287 | 996 056 005 | 1 004 311 271 | 37 294 444 |
| LZ4 | 0 | bitshuffle | 91.37 | 1 717 883 548 | 1 646 306 534 | 267 346 308 | 278 969 812 | 39 618 481 |
| LZ4FAST | 3 | shuffle | 91.78 | 453 166 592 | 463 579 997 | 1 013 467 965 | 990 702 418 | 37 753 332 |
| LZ4FAST | 3 | bitshuffle | 91.27 | 1 715 480 989 | 1 649 380 612 | 267 720 731 | 278 449 874 | 40 096 510 |
| LZ4FAST | 17 | shuffle | 91.64 | 451 584 784 | 457 452 644 | 1 017 017 934 | 1 003 972 389 | 38 406 860 |
| LZ4FAST | 17 | bitshuffle | 90.75 | 1 690 834 724 | 1 648 500 436 | 271 623 132 | 278 598 546 | 42 475 508 |
| *SL6* | -. | none | 92.19 | 254 777 476 | 173 152 489 | 1 802 631 187 | 2 652 400 937 | 35 870 622 |

**Table 4.** Average compression $C_r$, $\sigma$ and the coefficient of variation ($\widehat{C_v}$) obtained on the SWOT file obtained with the most efficient algorithms, ordered by the couple ($C_r$,$\sigma$) and illustrated in Figure 8

| Algorihthm | Average $C_r$ (%) | $\sigma$ (%) | $\widehat{C_v}$ |
|------------|-------------------|--------------|-----------------|
| LZ shuffle 9 | 76.98 | 34.31 | 0.45 |
| LZ bitshuffle | 76.46 | 35.18 | 0.46 |
| Zstd 22 | 74.83 | 38.05 | 0.51 |
| Zstd shuffle 1 | 78.95 | 32.98 | 0.42 |
| Zstd bitshuffle 1 | 77.12 | 34.06 | 0.46 |
| LZ4fast shuffle 17 | 75.71 | 35.36 | 0.47 |
| LZ4fast bitshuffle 17 | 75.42 | 34.80 | 0.46 |
| *SL6* polynomial | 75.33 | 33.93 | 0.45 |

is between 99.1 % (*SL6*) and 99.9 % (Zstd with shuffle). The other approaches are consistent in term of compression $C_r$ and standard deviation. The best result is obtained using `Zstd` associated to shuffle ($78.08; \sigma = 32.98$) and *SL6* ($75.33; \sigma = 33.93$). The Figure 9 illustrates the results obtained field by field. It is clear for the reader to see that they perform almost equally in terms of compression $C_r$, even if *SL6* tends to have a bit more homogeneous results than the others.

### 3.1.2 Compression speed

The average throughput results obtained for the SWOT file are quite heterogeneous from an algorithm to one another. They vary from 1.7 GB/s to only a few dozen MB/s for the slowest, the median equals to 238 MB/s. They are summarized in Table 5 and displayed in Figure 10. It can be seen that the Huffman algorithms associated with shuffle and bitshuffle filters perform quite well. For example, the LZ4Fast (parameter 17) associated with bitshuffle provides the lowest $\sigma$ but is relatively slow







**Figure 8.** Average compression $C_r$ (see Table 4 for the values) for the fields comprised in the SWOT file. It is clearly visible that all the algorithms perform well on this dataset, the lowest standard deviations are observerd for *SL6* and Zstd associated to the shuffle with. The Pithy algorithm provides good results on some fields too, unfortunately it is not able to compress some fields of the SWOT file.



**Figure 9.** Matrix of the compression $C_r$ of the fields of the SWOT file, it can be seen that the compression $C_r$ varies in a similar manner whichever algorithm is used, even if some are slightly better; the most interesting fields to look at are the one marked in red as they emphasize the differences between the algorithms.




**Table 5.** Average throughput (TP) ($MB/s$) and associated standard deviation $\sigma$ obtained on the SWOT file obtained with the most efficient algorithms, ordered by the couple ($C_r$,$\sigma$), results are plotted in the Figure 10.

| Algorithm | Average TP ($MB/s$) | $\sigma$ | $\widehat{C_v}$ |
|---|---|---|---|
| LZ shuffle 9 | 212.89 | 249.49 | 1.17 |
| LZ bitshuffle 9 | 120.24 | 203.09 | 1.69 |
| Zstd 22 | 80.75 | 67.12 | 0.83 |
| Zstd shuffle 1 | 833.98 | 344.46 | 0.41 |
| Zstd bitshuffle 1 | 225.78 | 54.33 | 0.24 |
| LZ4Fast shuffle 17 | 965.33 | 231.47 | 0.24 |
| LZ4Fast bitshuffle 17 | 254.47 | 21.59 | 0.08 |
| *SL6* polynomial | 1431.24 | 753.44 | 0.53 |

(254.46 MB/s). Indeed, it is outperformed by the same with the shuffle preconditioner. The *SL6* algorithm is 48 % faster than the fastest Huffman-based, even when it is associated to shuffle and bitshuffle filters. The LZ4Fast associated to the bitshuffle has the most homogeneous results for the SWOT dataset.

A closer look at the results obtained for the SWOT `pixel_area`, in Figure 11 field shows more variability in the compression throughput, depending on the algorithms. We choose to display this field in particular since it emphasizes the disparity of 270 the results. Here, the *SL6* algorithms ranks first and second best performance (more than 2.5 GB/s for 48.3 % compression $C_r$ for the first), the second has a higher compression $C_r$ (76.5 %) for 1.75 GB/s.

Since it is a stream algorithm, the compression throughput of *SL6* is constant for a given word size, moreless a few percent. The throughput for 8-bit is the lowest 329 MB/s, while the figures on 64-bit data is 1817 MB/s, 32-bit data words lead to 1008 MB/s of throughput. This is easily understandable by the fully deterministic nature of the algorithm and the fact that the 275 algorithm locally compress the data using blocks of predetermined size.

### 3.1.3 *H*-score

The *H*-score defined in the previous section emphasize the algorithms that provide consistent compression $C_r$ and consistent throughput. This is extremely important when the dataset can be heterogeneous and when the targeted application is real-time compression on a single-core. Since *SL6* provides quite high throughput with a compression $C_r$ comparable to the other 280 algorithms – and for both a reasonable $\sigma$ – it scores higher than the others. It is followed by LZ4Fast associated to the shuffle filter (parameter 17), which scores 2 % lower than the *SL6* algorithm. The results are visible in Table 6 and in Figure 12.

### 3.2 Threshold Throughput

The results for the transmission-threshold-throughput ($tt_{tx}$), that was depicted in subsection 2.4 is considered in this subsection. When it is higher than the media throughput, the time used to compress the data, to transmit and to decompress it is lower than 285 the time required for the transmission of the original data. This metric is highly correlated with the speed of the hardware used







**Figure 10.** Plot of the average compression throughputs for the `swot` fields, the standard deviation is given as error bars, on *SL6* the errors bars are quite important since we mixed words of data of different size (8, 32 and 64 bits). The errors bars for a given datasize (for example 64 bits) is constant with minimal standard deviation.







**Figure 11.** Plot of the decompression speeds for the most efficient algorithms for the SWOT `pixel_area` field, the compression $C_r$ are given as labels on the top of each histogram boxes, some of the algorithms provides good results with different parameters and preprocessing (ZStd and LZ4 Fast); the Pithy algorithm has been excluded since it is not able to compress some other fields of the file. SLx was tested with different blocks sizes.





**Table 6.** *H*-score calculated on all the fields of the SWOT file. Since *SL6* has the highest throughput and is among the most homogeneous results, it outperforms the others approaches.

| Algorithm | *H*-score |
|---|---:|
| LZ shuffle | −4.62 |
| LZ bitshuffle | −9.53 |
| Zstd 22 | 0.92 |
| Zstd shuffle 1 | 35.65 |
| Zstd bitshuffle 1 | 11.02 |
| LZ4Fast shuffle 17 | 44.92 |
| LZ4Fast bitshuffle 17 | 13.96 |
| *SL6* polynomial | 54.52 |



**Figure 12.** Plot of the H-scores calculated on the results for the SWOT file.





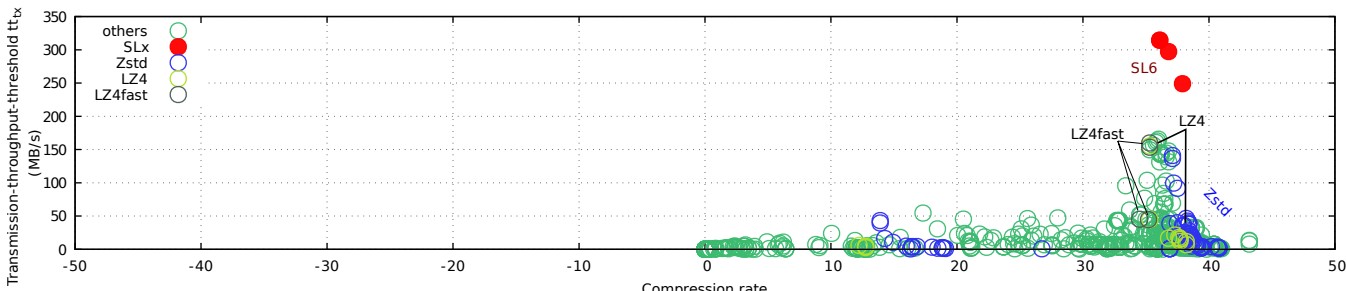

**Figure 13.** Ranking of the algorithms, $tt_tw$ (y) vs. the compression $C_r$ (x) for the SWOT `latitude` field.

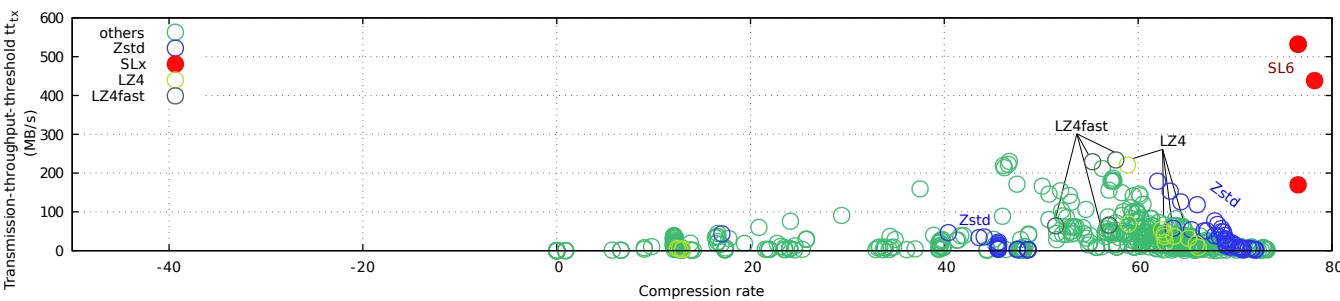

**Figure 14.** Ranking of the algorithms, $tt_tw$ (y) vs. the compression $C_r$ (x) for the SWOT `L2 pixel_area`.

to execute the algorithms. Here, since the hardware is the same for all of them, these calculated metrics are comparable. Of course, the higher is better, it means that even with a extremely fast transmission channel, compressing and uncompressing the data on-the-fly is worth it. For example, on the Figure 13, it can be seen the $tt_{tx}$ of the *SL6* algorithm (in red) is higher than 300 MB/s which corresponds to more than 2.4 Gb/s. Transmitting on a media that has a throughput lower than 2.4 Gb/s

would benefit from compression using this algorithm. This is the combination of a good compression $C_r$ (around 40 %) (X axis) and a low computation time. If now the media throughput is 120 GB/s (1 Gb/s), then using all the algorithm above this threshold would be more efficient than transmitting the original data, it means roughly the best LZ4fast LZ4 and Zstd variants, a few others one and *SL6*. As the compression time depends on the data, the, we have plotted the $tt_{tx}$ for the pixel_area field in Figure 14. As reminder, the pixel_area field is quite smooth which makes it easier for the algorithms like the *SL6* which

performs mathematical signal processing. Here, for the *SL6* algorithm, the $tt_{tx}$ is higher than 500 MB/s which corresponds to more than 4 Gb/s for a compression $C_r$ of more than 75 %.

## 3.3 Memory Usage

Memory usage of all tested algorithms is lower than 2 MB as well as for compression and decompression. Because of the necessity of building a dictionary, the Huffman-based ones often reach 2 MB. Their memory usage is not deterministic and

depends on the data. The lowest is *SL6* with less than 128 kB (for the 64-bit floating point) version. Most of the memory usage





**Table 7.** Average results for the fields from the SWOT file that are the most difficult to compress, the compression $C_r$ and the standard deviation, the compression and decompression throughput and the H-score are displayed.

| Algorithm | Rate | $\sigma$ | Comp. tp | $\sigma$ | Decomp. tp | $\sigma$ | H-score |
|---|---|---|---|---|---|---|---|
| | % | % | $MB/s$ | $MB/s$ | $MB/s$ | $MB/s$ | |
| LZ bitshuffle 9 | 28.46 | 23.27 | 180.41 | 5.68 | 24.35 | 71.17 | 0.19 |
| Zstd 22 | 21.35 | 27.03 | 4.51 | 0.95 | 768.78 | 580.69 | −0.18 |
| Zstd shuffle 1 | 31.96 | 23.77 | 371.82 | 109.61 | 741.43 | 117.69 | 19.46 |
| Zstd bitshuffle 1 | 28.63 | 23.51 | 155.22 | 14.21 | 264.38 | 19.56 | 6.54 |
| LZ4 shuffle | 25.97 | 24.85 | 604.12 | 158.14 | 1009.12 | 64.52 | 4.51 |
| LZ4Fast shuffle 17 | 23.82 | 25.03 | 741.24 | 150.69 | 1028.61 | 50.26 | −6.49 |
| LZ4 bitshuffle 0 | 27.26 | 22.50 | 217.77 | 7.02 | 278.56 | 4.56 | 9.10 |
| *SL6* polynomial | 24.35 | 15.74 | 540.86 | 127.92 | 1320.52 | 230.79 | 32.21 |

is used for the input and output buffers. Another key point is that the *SL6* memory usage is constant and does not depend on the data but only of the size of the words (8, 16, 24, 32, 64 bits), while the choice of the parameters has a limited impact on the memory usage of less than 5 %. Alternatively, the fact the Huffman based methods have to construct a dictionary requires memory, and makes embedded implementation bit more complex, espcially on hardware targets. This is usually cope by
working on chunks of data, in that cas, the compression level can be lower. Consequently, this feature makes the *SL6* approach ideal for embedded systems.

### 3.4 Results on relatively complex fields

The entropy of most of the fields of the SWOT file is quite low, which explains the high compression $C_r$. We decided in this subsection to focus on fields for which algorithms have more difficulties to compress. The average results are displayed in
Table 7. The ranking differs from Tables 5 and 6. Indeed, for example, the LZ4Fast shuffle has a low $H$-score because of the important variability in the compression $C_r$: some of the fields are almost not compressed while the LZ4 bitshuffle now appears in the ranking at the third position. Figures 15 and 16 show that the compression $C_r$ of the *SL6* algorithm is comparable to the other Huffman-based approaches associated to shuffle and bitshuffle filter. However, it shows superior performances in terms of compression speed. The high score of *SL6* is mainly due to the consistent compression $C_r$ among the dataset associated
to one of the best throughput ($3^{rd}$). Some of the fields are (almost) not compressed by the LZ4 shuffle and LZ4Fast shuffle. Consequently, these algorithms tend to copy portions of the fields without any modification, leading to a quite high throughput.

### 4 Discussion

The results obtained with the extended testbench performed in this work show that the state-of-the-art algorithms performs
similarly in term of compression rate. Indeed, if there is a certain level of variability among the different fields of the data





**Figure 15.** Plot of the compression speeds over 300 GB/s, as histogram boxes, and the associated $C_r$ as blue points, for the SWOT `Pixel_Area` field.

acquired by the swot mission, the average rate for the best algorithms is comprised between 75 % and 82 %. Considering that metric only, the LZ algorithm associated to bitshuffle (parameter 9) and Zstd associated to a shuffle preprocessing (parameter 1) provide the best results. However, the worst one does less than 10 % lower. Thus, the compression rate only is not a metric to be considered provided that we pick the algorithm among the top ten best compression rate. It is worth to note that some

algorithms are not able compress some fields, this is the case of Pithy, some variants of the LZ4 etc. They have been excluded from the final results, even if for some fields they provide good performances typically the Pithy algorithm associated to the shuffle filter as visible on the Figures 15 and 16.

An other metric that needs to be considered are the memory usage of the algorithms performing compression and decompression. This information is not really relevant if the algorithms are executed by a workstation that can holds several GB of





**Figure 16.** Plot of the compression speeds over 500 MB/s, as histogram boxes, and the associated $C_r$, as blue points, for the SWOT `CrossTrack` field.

Random Access Memory (RAM), but it is extremely relevant if they are executed by an embedded system which can be quite limited. Moreover, for space application, memory are prone to faults since cosmic rays can deposit their energy into one or several cells. Thus choosing algorithms that have a low memory usage for such implementations make sense. The *SL6* algorithm has a memory footprint lower than 128 kB while the others vary between 1 MB and 2 MB, which makes it a good candidate for embedded systems. This is also a good indicator of the capacity of the algorithm to be implemented in an embedded programmable microcontroller or hardwired for a Field-Programmable Gate Array (FPGA) implementation.

The couple compression rate and throughput is a good way to rank the algorithms. However, since the analysis of the results we have obtained have shown a quite important variability in the results, we decided to propose the $H$-score to take it into account into a combined metric. It shows that the top-3 algorithms are the *SL6* ($> 50$), the LZ4Fast associated ot shuffle ($> 40$) and the *ZStd* associated to shuffle which scores higher than 35.





Another interesting point is that the compression throughput of *SL6* is constant for a given datatype, which is quite understandable as it operates in stream processing.

Beyond the choice of a compression algorithm for space-to-ground communication, where constraints require approaches that can be easily embedded and have consistent compression and throughput performances, the ground applications related to the SWOT mission is also considered. Compression for ground applications is extremely important too as the SWOT mission

may generate around 5 TB of data per day. Indeed, ultimately, the data will be used in numerical models and computed on HPCs, thus, it is important to be able to compress and decompress the messages on-the-fly without any loss of performance. Compression of HPC messages is a good way to reduce computing time, provided that the compression/decompression is fast enough, that is the reason why we have introduced the transmission-throughput-threshold. In the same vein, to enable smooth storing an manipulating of these data requires, fast compression algorithms are required, ideally that can perform on-the-fly,

with no or almost no perceptible delay for the operators. The *SL6* approach is the only one which has constant compression time for a given datawidth (64, 32, 16 or 8-bit words). It is also able to compress a signal with fixed-size buffers, which participates to its deterministic characteristics in terms of compression time and hardware resources that are required for the *SL6* algorithm execution.

Another key point is the determinism of the compression rate, indeed, the storage and database systems will be sized for an

expected volume of data. The H-score metric that is proposed this paper clearly emphasizes the properties that we are seeking for both space and ground application.

## 5   Conclusion

Since space missions are generating more and more data, for example 7 PB is expected for the swot mission, efficient data compression approaches needs to be explored. Huffman-based algorithms are traditionally used for scientific application, es-

pecially for netCFD data. It has been shown in the past they are quite efficient especially when associated to to filters such as shuffle and bitshuffle. This paper compared Huffman based algorithms and the *SL6* approach. The Huffman-based algorithms require memory (shuffle needs to analyze the entire file) and their performances are heterogenous, strongly depending on the data. When associated to shuffle filter, stream processing is not possible. Alternatively, the *SL6* algorithm provides homogeneous results, low memory usage, constant compression time for a given datatype, especially in terms of compression $C_r$. On

the dataset that is representative of the SWOT mission, the compression throughput is 35 % faster than the best algorithm of the bench for a similare compression level. It is also often almost twice faster than the other algorithms of this extensive bench.

*Code availability.*   The code of the testbench was developped in Python and C. It uses the LZBench available on https://github.com/inikep/
lzbench. The code for **SLX!** (**SLX!**) can be provided by Subnet, Pigoury@Subnet.fr. The codes for the other compression algorithms are
included in the LZBench as opensource.



*Data availability.* The SWOT dataset is provided by CNES and is available upon demand.

*Author contributions.* OT is the inventor of the *SL6* compression algorithm, OT performed the benchmark, SP and MT have done the data analysis, FG provided the swot data and the comparison methodology, MT and FG wrote the paper

*Competing interests.* The *SL6* algorithm is under license at Subnet SAS – http://www.subnet.fr.

*Acknowledgements.* The authors would like to thank the Centre National des Etudes Spatiales (CNES) for funding this research and Subnet
for providing the sources of the *SL6* algorithm. They also would like to thanks Nicola MARTIN for the proofreading work.



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
