# Peer review of "A Comparison of Lossless Compression Algorithms for Altimeter Data"

_EGUsphere, 2022_

## Referee Comment (RC1)

**Comments**

In this manuscript, the authors compared Huffman-based algorithms and the SL6 approach in processing data from the SWOT mission. The topic of the manuscript is important and also interesting. However, I do feel that the paper fails in many aspects such as the reproducibility of the study and the writing. Thus, I suggested the authors make a deep revision on this manuscript. The detailed comments are listed below.

1. After reading the paper, I got the general feeling that the SL6 approach is better than other approaches in processing the data given in this study. However, the source code of SL6 is not open-sourced, which means the results shown in this paper cannot be reproduced by others. As has been commented by the chief editor, I also do not think it is permitted by the GMD journal, as the aim of the journal is to share new models and new methods to others and the public.

2. The abstract lacks the description of the key findings achieved in this study.

3. In some places of the manuscript, the style of the reference citation is strange. For example, L18, (Sudmanns et al. (2020)), (Elsey (1968)); L72, (Diane Poirie and et al. (1998); Christopher Rumsey, Bruce Wedan and Poinot (2012)) implementing Advanced Data Format (ADF) (Owen

and Daniel (1998)) and HDF5 (Folk et al. (2011)).

4. In section "Introduction", what about the conclusions of previous studies? Has someone made similar comparisons? It is always better to state the differences between your research and others in the introduction section.

5. Again, the dataset of SWOT mission is also not publicly available, which limits the reproducibility of the study.

6. The writing of the manuscript needs to be substantially improved. I just list some of them below.

7. L13, Satellites for Earth observations is

8. L14, this topics

9. L26, is usually application

10. L37, Welton et al. (2011)) in the tools for the manipulation of the HDF5 formats of the Computational Fluid Dynamics (CFD) General Notation System (CGNS) and benchmarked by previous works like in Delaunay et al. (2019); I did not find the verb in this sentence.

11. L68, developped

12. L90, generatedd

13. L115, are use

14. L173, do not use asterisk to replace the multiplication symbol.

15. L192, H – score in the equation looks like H minus score. I suggested to change the symbol H-scrore to H_score.

16. L220, most interesting ones of them

17. Figure 7, I am not sure about the meaning showing the figure here, as the readers cannot use it anyway.

18. L305, in that cas

19. L321, swot -> SWOT

20. L328, An other -> Another

21. L355, is proposed in this paper

22. L358, swot -> SWOT

23. L360, to